# A Colorimetric Strategy Based on Aptamer-Catalyzed Hairpin Assembly for the On-Site Detection of *Salmonella typhimurium* in Milk

**DOI:** 10.3390/foods10112539

**Published:** 2021-10-22

**Authors:** Sihan Chen, Xinran Zong, Jiapeng Zheng, Jiaxin Zhang, Mengyao Zhou, Qing Chen, Chaoxin Man, Yujun Jiang

**Affiliations:** Key Laboratory of Dairy Science, Ministry of Education, Department of Food Science, Northeast Agricultural University, Harbin 150030, China; sihan_chen@ahau.edu.cn (S.C.); zongxinran@126.com (X.Z.); zjp1997447@163.com (J.Z.); jiaxin1998Z@163.com (J.Z.); zhoumengyao1997@163.com (M.Z.); chenqingchen163@163.com (Q.C.); cxman@neau.edu.cn (C.M.)

**Keywords:** *Salmonella typhimurium*, catalytic hairpin assembly, AuNPs, on-site detection, aptamer

## Abstract

*Salmonella typhimurium* (*S. typhimurium*) is a foodborne pathogen that has caused numerous outbreaks worldwide, necessitating the development of on-site strategy to prevent early contamination. Here, we set up an enzyme-free strategy for aptamer-catalyzed hairpin assembly in which salt-induced aggregation of unmodified gold nanoparticles (AuNPs) served as a colorimetric signal output, allowing on-site detection of *S. typhimurium* in milk. The aptamer-functionalized magnetic beads were used as a vehicle of specifically enriching target bacteria which conjugated with target aptamer to trigger the “Y” shape catalytic hairpin assembly (Y-CHA) circuit. Due to the hairpins desorbing from the surface of AuNPs to the formation of a large amount of double-stranded DNA (dsDNA), AuNPs turned from dispersion to aggregation in the presence of *S. typhimurium*, resulting in a change of the colorimetric signal from red to blue-gray. The signal output showed a linear relationship for *S. typhimurium* over a concentration range of 10^2^ to 10^6^ CFU/mL, with a sensitivity of 2.4 × 10^2^ CFU/mL under optimal conditions. The visual protocol has excellent selectivity even in the presence of other competitive bacteria and has been validated in real milk samples with a sensitivity of 2.8 × 10^3^ CFU/mL.

## 1. Introduction

As a foodborne pathogen that spreads widely between animals and humans, *Salmonella typhimurium* (*S. typhimurium*) is one of the most frequently reported causes of food poisoning and acute intestinal diseases, contributing to a major impact on global food safety [1,2,3]. Dairy products are one of the main routes for transmission of *S. typhimurium* [4,5]. Conventional culture methods for the detection of *S. typhimurium* in food samples rely on enrichment steps, plating on selective culture media, and biochemical and/or serological identification, which often suffer from being time-consuming and laborious, resulting in unsuitability for products with short shelf life such as milk [6,7,8]. For molecular methods such as PCR, their enzyme-dependent properties bring about sophisticated operations and high cost, making it difficult to be effectively promoted in economically underdeveloped regions [9,10]. Therefore, the development of novel on-site approaches is necessary for detecting *S. typhimurium* in milk.

In order to overcome the limitations of enzymatic amplification, including complex operations, specific reaction conditions, and reaction time-dependent enzymatic activity, a novel signal amplification method, catalytic hairpin assembly (CHA), has been proposed in recent years. CHA is a strategy that depended on self-assembly and disassembly reactions between nucleic acid hairpins to achieve signal amplification, broadening new horizons for on-site detection [11,12]. Compared with other enzyme-free amplification methods, CHA circuit has higher catalytic efficiency, lower background signal, and a simpler reaction system [13,14]. In typical CHA, two metastable hairpin DNA monomers are activated by the initiation strand and driven by free energy, which sequentially self-assemble and displace the initiator to generate a stable linear double-stranded DNA (dsDNA) structure [15]. To further improve the sensitivity and stability of the assay, “Y” shape CHA (Y-CHA) has emerged as a powerful tool because of its high payload efficiency and excellent biostability [16]. The researchers confirmed that the detection time of Y-CHA was shortened by 1.5 h compared with one-dimensional CHA, because the more stable “Y” shape DNA structure can promote the strand displacement reaction between hairpins [17].

Gel electrophoresis is the primary approach to the characterization of CHA products; however, its dependence on lengthy analysis procedures and toxic reagents has restricted the application of CHA in on-site assays [18]. Thanks to the excellent biocompatibility and perfect signal performance, CHA can be easily coupled with gold nanoparticles (AuNPs), enabling simple, low-cost, and visual observation of amplification products [19,20]. With the unique optical properties of high molar extinction coefficients and local surface plasmon resonance (LSPR), AuNPs exhibit a red to blue-gray color change owing to the slight change of interparticle distance [21,22,23]. The chromogenic response of AuNPs can be demonstrated by the different adsorption behavior of single-stranded DNA (ssDNA) and dsDNA toward unmodified AuNPs. Since ssDNA can adsorb to AuNPs via van der Waals attraction between exposed bases and nanoparticles, the electrostatic repulsion between AuNPs is enhanced thereby avoiding salt-induced AuNPs aggregation [17,24]. In contrast, there is negligible binding between dsDNA and negatively charged AuNPs under strong repulsive forces, due to the exposed negatively charged phosphate skeleton of dsDNA, resulting in self-aggregation and color change of AuNPs upon salt induction [25,26]. Therefore, the concentration of target can be easily observed with the naked eye, making the assay free from reliance on complicated instruments.

Herein, we developed an aptamer-catalyzed hairpin assembly signal amplification strategy through combining the recognition and programmability of aptamers and applied to the on-site colorimetric detection of *S. typhimurium* in milk. In the work, two anti-*S. typhimurium* aptamers were acted as biorecognition elements, conjugated with magnetic beads to specifically enrich *S. typhimurium* in milk and form the sandwich complexes. One of the aptamers acted as ssDNA to trigger the Y-CHA circuit, leading to AuNPs aggregation and red-shifting to blue-gray in the presence of salt. This strategy eliminated the need for complex enzymatic steps, expensive instruments and experienced operators, and therefore, promises to be a powerful tool for on-site detection of *S. typhimurium*.

## 2. Materials and Methods

### 2.1. Materials and Reagents

Streptavidin magnetic beads (SMBs) (1.0 μm in diameter, 10 mg/mL) were purchased from Invitrogen Biotechnology Co., Ltd. (Carlsbad, CA, USA). Salmon sperm DNA and yeast tRNA were both obtained from Solarbio Tech Co., Ltd. (Beijing, China). The sequences of hairpins were elaborately designed by NUPACK software (http://www.nupack.org/, accessed on 2 October 2021) (Appendix A) [27]. Two anti-*S. typhimurium* aptamers, one of which was modified by biotin as a capture aptamer (Bapt), and the other as a target aptamer (Tapt), were screened in the literature [28]. All DNA sequences used in this study were displayed in Appendix A, synthesized and purified by Sangon Biotech Co., Ltd. (Shanghai, China). All reagents and solvents used in this study were of analytical grade and utilized without further purification, and Milli-Q water (Millipore, Burlington, MA, USA, 18.2 MΩ cm) (Molsheim, France) was used in all the works.

The UV–Vis absorbance spectra were recorded with a UV-2600 spectrophotometer (Shimadzu Co., Ltd., Tokyo, Japan) with Quartz cuvettes. Transmission electron microscopy (TEM) images were supplied from Hitachi, Co., Ltd. (Tokyo, Japan).

### 2.2. S. typhimurium Culture 

*S. typhimurium* (ATCC 14028) was used as a model target strain for the enzyme-free signal amplification strategy. Other pathogenic microorganisms such as *Staphylococcus aureus* (*S. aureus*) ATCC 25923, *Staphylococcus epidermidis* (*S. epidermidis*) ATCC 12228, *Escherichia coli* (*E. coli*) ATCC 25922, *Cronobacter sakazakii* (*C. sakazakii*) ATCC 29544, *Cronobacter malonaticus* (*C. malonaticus*) DSM 18702, *L. monocytogenes* ATCC 19114, *Listeria welshimeri* (*L. welshimeri*) ATCC 43550, *Vibrio parahaemolyticus* (*V. parahaemolyticus*) ATCC 17802, *Shigella Flexner* (*S. Flexner*) CMCC 51572, *Bacillus cereus* (*B. cereus*) CMCC 63303, *Enterobacter aerogenes* (*E. aerogenes*) ATCC 13048, and *Enterobacter cloacae* (*E. cloacae*) CMCC 45301 were used for the validation of the colorimetric assay. All bacteria were grown in lysogeny broth (LB) with gentle shaking (150 rpm) at 37 °C for 12 h, and the number of cells was determined by plate counting on LB agar. The bacterial cells were harvested by centrifugation (5000× *g*, 5 min) and resuspended in sterilized phosphate-buffered saline (PBS), which allowed to remove the metabolites accumulated in the medium which may adversely affect the colorimetric analysis.

### 2.3. Preparation of AuNPs

The AuNPs were synthesized by the citrate reduction of HAuCl_4_ following the literature procedures with improvements [29]. In brief, 100 mL solution containing 0.01 g HAuCl_4_ was heated to its boiling point while stirring. Then, 1 mL of 1.0% trisodium citrate solution was added to the boiling solution quickly, resulting in a change in solution color from pale yellow to wine red. The mixture solution was kept boiling for 15 min and then cooled to room temperature. The product was stored in a dark glass bottle at 4 °C for further use.

### 2.4. Y-CHA Circuit and Gel Electrophoresis

Before constructing the Y-CHA circuit, all of the hairpins (Y1, Y2, and Y3) were denatured at 95 °C for 5 min and dropped 5 °C per minute until cooling down to room temperature [12]. The 100 nM Tapt was mixed with 1 μM of each hairpin in the sterilized TNaK buffer (20 mM Tris, 200 mM NaCl, and 5.0 mM KCl) and performed at 37 °C for 2 h. Then, 5 μL of the sample was mixed with 1 μL of loading buffer and were then run through 4% agarose gel electrophoresis (AGE) in 1 × TBE buffer (90 mM Tris-HCl, 90 mM boric acid, 2 mM EDTA, pH 8.0) at 160 V for 60 min. The gel was visualized via AlphaImager HP imaging system (Alpha Innotech Crop., San Jose, CA, USA).

### 2.5. Synthesis of the Capture Complexes

SMBs conjugated with Bapt were synthesized following a previously reported protocol [30]. In brief, the 12 μL of SMBs were washed with an equal volume of 1 × B&W buffer (5 mM Tris-HCl, 500 nM EDTA, 1 M NaCl). The solution was then mixed with 10 μL of Bapt (2 μM) and incubated at room temperature for 10 min. Afterwards, the supernatant was removed, and the pellet was resuspended in 100 μL of PBS containing salmon sperm DNA (1 mg/mL) and yeast tRNA (0.1 mg/mL) for 10 min. Then, the capture complexes (SMBs-Bapt) were collected after washing with PBS and maintained at 4 °C for further use.

### 2.6. Colorimetric Detection of S. typhimurium

The 10 μL of 100 nM Tapt was first incubated with the bacterial suspension for 30 min, and then incubated with the SMBs-Bapt for 30 min at 37 °C. The residual Tapt was removed by washing with PBST. Subsequently, the SMBs-Bapt-*S. typhimurium*-Tapt conjugates were obtained by resuspending the precipitate in 10 μL of ultra-pure water. Tapt was released by heating at 60 °C for 5 min and transferred to a TNaK solution containing 100 nM Y1, Y2, and Y3. After incubation at 37 °C for 2 h, 10 μL of Y-CHA products were mixed with 100 μL of AuNPs, incubated for 2 min and photographed to visualize the color intensity change. The absorption spectrum of the solution was obtained over a wavelength range of 520–650 nm using a UV–Vis spectrometer.

### 2.7. Detection of S. typhimurium in Spiked Milk

The practicability of the biosensor was evaluated with *S. typhimurium* spiked into the commercially available UHT milk. With reference to our previous work, the spiked milk samples were pretreated as follows [30]. Briefly, 1 mL of overnight activated *S. typhimurium* cultures was diluted and added to 24 mL of each sample with a final concentration of 2.8 × 10^8^ to 2.8 × 10^1^ CFU/mL. Then, 1 mL of the mixture was centrifuged at 5000× *g* for 5 min and resuspended in PBS solution to remove the endogenous protein of spiked milk samples. The obtained skim milk samples were tested by the constructed Y-CHA colorimetric aptasensor, and all analyses were repeated in triplicate. The milk sample without *S. typhimurium* was considered as experimental control.

## 3. Results and Discussion

### 3.1. Principle of the Strategy

The design principle for the on-site colorimetric strategy based on Y-CHA for the detection of *S. typhimurium* is schematically illustrated in Figure 1. First, the two aptamers were specifically conjugated with *S. typhimurium* and integrated the SMBs-Bapt-*S. typhimurium*-Tapt sandwich complexes. Upon heating, Tapt was released and hybridized with the sticky end (domain d*c*b*a*) of Y1 via a branching migration reaction, resulting in the formation of Tapt-Y1 intermediate. The newly exposed single-stranded sequences (domain b a e a) of Tapt-Y1 further hybridized to the sticky end (domain a*e*a*b*) of Y2 to generate Tapt-Y1-Y2 intermediate. Analogously, the exposed sequences of Tapt-Y1-Y2 continued to open hairpin Y3 and displaced the initiator (Tapt) to promote a stable Y1-Y2-Y3 “Y”-branched structure. The replaced Tapt continued to trigger the next round of the Y-CHA circuit until all the hairpins were exhausted. Therefore, the hairpin probes fell off from the AuNPs surface, causing AuNPs lost the protection of hairpins and underwent self-aggregate in the salt solution, making a change in color from red to blue-gray. In contrast, when the target pathogen was absent, hairpins Y1, Y2, and Y3 were kinetically unable to open their stem-loop structures spontaneously as the melting temperatures of hairpins (Y1 = 83.9 °C, Y2 = 85.4 °C and Y3 = 80.0 °C), calculated by mfold software (http://www.unafold.org/mfold/, accessed on 2 October 2021), were much higher than the reaction temperature. Thus, AuNPs were attracted by the sticky end of the hairpins under electrostatic attraction and dispersed stably in the salt solution with red color [24].

TEM and UV–Vis spectroscopy were performed to prove the viability of our strategy. Appendix A characterized the TEM images of the biotin-modified anti-*S. typhimurium* aptamer conjugated with SMBs, indicating that the prepared aptamer-functionalized magnetic beads can bind tightly to the target microorganisms. As shown in Figure 2A, a maximum absorption peak of AuNPs was observed in the spectrum at approximate 520 nm and solution color was red without the initiator Tapt. In contrast, the addition of Tapt significantly induced an increase at 650 nm, and a decrease at 520 nm, with the solution turning blue-gray. As a result, the absorbance ratios of A650/A520 were used for quantitative analysis. The TEM images indicated that the AuNPs exhibited spherical morphology with a diameter of approximately 15 nm and dispersed well in the field of view without Tapt (Figure 2B). While in the presence of the target DNA, the AuNPs were bound together (Figure 2C). The TEM results were consistent with UV–Vis absorption spectra as well as color changes.

### 3.2. Gel Electrophoresis Analysis

AGE analysis was performed to investigate the Tapt-triggered Y-CHA circuit, in which hairpins Y1, Y2, and Y3 were denatured before the reaction to dissociate intermolecular interactions and form hairpin structures. As shown in Figure 3, one clear single band in each of lanes 1–4 was Tapt, Y1, Y2, and Y3, respectively. No new bands were generated in the absence of the initiator Tapt (lane 5), indicating that the mixture of the three probes maintained sub-stable hairpin structure. However, when Tapt was incubated with the hairpins, the original bands of the individual hairpins could be observed to roughly disappear and a new band appeared at around 160 bp (lane 6), indicating that the Tapt-triggered Y-CHA circuit successfully occurred. 

### 3.3. Feasibility and Mechanism of AuNPs Colorimetric Detection

To explore the feasibility and mechanism of AuNPs against salt-induced aggregation, the AuNPs were incubated with different hairpin probes. Sample 1 was AuNPs without hairpin probes; sample 2 was AuNPs with hairpins Y1, Y2, and Y3; sample 3 was AuNPs with Y1, Y2, and Y3 in salt solution; sample 4 was AuNPs with Y1, Y2, Y3, and Tapt in salt solution; and sample 5 was AuNPs with the blunt end hairpins Y1*, Y2*, and Y3* in salt solution. The A650/A520 ratios for each sample were shown in the bar graph of Figure 4. Samples 2 and 3 displayed the same color as the original AuNPs (sample 1), demonstrating that the hairpin probes with sticky ends adsorbed on the surface of AuNPs to stabilize them under salt induction. In sample 4, the hairpins self-assembly occurred to form branching dsDNA products in the presence of the initiator Tapt, resulting in the aggregation of AuNPs with a blue-gray color. Sample 5 also showed blue-gray color but its A650/520 was slightly lower than that of sample 4, which was probably due to the flexibility of the loop domain as single-strand structure, which could bind to AuNPs and reduce to some degree the aggregation of AuNPs. Therefore, the results confirmed the feasibility of visual detection of Y-CHA products by unmodified AuNPs.

### 3.4. Finding the Optimum Conditions

The performance of the assay is influenced by parameters such as concentration of NaCl, concentration of hairpins, incubation temperature and incubation time. These parameters should be optimized to ensure rapid and accurate colorimetric analysis for *S. typhimurium* concentrations. Salts serve the purpose of affecting the stability of hairpins and Y-CHA intermediates by neutralizing the negative charge on the DNA backbones in order to reduce the repulsion between phosphates [31,32]. Meanwhile, the salt concentration in the solution plays a key role in the degree of aggregation of AuNPs. As shown in Figure 5A, the degree of AuNPs aggregation in the positive samples peaked when the NaCl concentration was 200 mM and the stability of AuNPs in the negative samples was not destroyed. The concentration of the hairpin also plays an important role in the colorimetric process. When the hairpin concentration was 100 nM, the AuNPs kept stable dispersion in the negative sample (Figure 5B). While the hairpin concentration continued to increase, the excess hairpins that were not exhausted by the Y-CHA circuit would adsorb on the surface of AuNPs, leading to a reduction in the aggregation of AuNPs in the positive samples, which considerably reduced the sensitivity of the proposed method [33].

The incubation temperature is an important factor affecting the stability of hairpins secondary structure and intermediates. Figure 5C showed that the reaction rate of Y-CHA increased with increasing temperature, due to the enhanced free energy and collision chance between hairpins. However, at temperatures above 37 °C, an increase in the A650/A520 value of the negative sample also occurred, producing a higher background signal. The incubation time directly affects the generation of Y-CHA products, which has an impact on the colorimetric performance of AuNPs. Figure 5D recorded the colorimetric results of A650/A520 as a function of incubation time. The value of A650/A520 maintained its increase with assembly time and reached a plateau at 120 min. As time continued to extend, the hairpins in the positive samples had been largely consumed completely, allowing the A620/A520 values to be nearly constant. However, for negative samples, the longer the reaction time, the higher the chance of hairpin “leakage”, as the probes did not fully form the ideal stem-loop structure during annealing, resulting in a low amount of dsDNA even in the absence of the initiator strand and leading to high background noise.

### 3.5. Analytical Performance of the Proposed Strategy

Under optimized conditions, the sensitivity of the on-site colorimetric strategy based on Y-CHA was evaluated by visual observation and measurement of absorption spectra. Figure 6A presented the color changes of AuNPs induced by magnetic bead enrichment and Y-CHA signal amplification for *S. typhimurium* with concentrations ranging from 2.4 × 10^1^ to 2.4 × 10^8^ CFU/mL. Samples containing *S. typhimurium* in the concentration range of 2.4 × 10^2^ to 2.4 × 10^8^ CFU/mL were successfully amplified, which induced the aggregation degree of AuNPs making the color gradually change to blue-gray. In the spectrogram of Figure 6B, the absorbance of the AuNPs solution at 520 nm decreased successively with increasing *S. typhimurium*, while the absorbance increased to around 650 nm. The correlation curve was constructed based on the absorbance peak ratio at 650 nm and 520 nm (A650/A520) versus the concentration of bacteria. It can be observed from Figure 6C that a good linearity in the range from 10^2^ to 10^6^ CFU/mL (y = 0.21196, x = 0.21667, R^2^ = 0.98297).

To compare the amplification efficiency of Y-CHA system, we designed two hairpins (C1 and C2) to carry out a conventional one-dimensional CHA circuit. The incubation time of the traditional CHA circuit was increased by 30 min over Y-CHA (see Appendix A). We used the traditional CHA circuit for the detection of *S. typhimurium*, and it could be noticed in Appendix A that the color of the AuNPs solution only changed slightly when the cell concentration was 5.2 × 10^3^ CFU/mL. The results suggested that the sensitivity of Y-CHA was 20-fold higher than that of conventional CHA, possibly caused by the different stability of the Y-shaped DNA structure formed by three hairpins versus the linear DNA structure formed by two hairpins. The Y-CHA circuit offered extra toeholds and formed more nucleic acid assemblies by opening Y1, Y2, and Y2 to improve reaction efficiency. Moreover, the stem of “Y”-shaped dsDNA produced by Y-CHA is 120 bp in length, which is more stable than the linear dsDNA with 60 bp, thus facilitating the reaction and reducing the detection time.

To investigate the specificity of this colorimetric strategy for *S. typhimurium* detection, interference experiments were performed with several common foodborne pathogens and heat-treated *S. typhimurium* cells. As shown in Figure 7, for non-target and dead *S. typhimurium*, the color of AuNPs solution was red, which was significantly different compared with positive samples (blue-gray). In addition, the absorbance data of the samples also showed significantly lower values of A650/A520 for non-target bacteria than for live *S. typhimurium*. The assay distinguished between dead and viable cells mainly due to the degradation of outer membrane proteins of *S. typhimurium* caused by heat treatment, which prevented the bacteria from binding to the aptamer [34]. These results demonstrated that our proposed strategy has considerable sensitivity and selectivity to *S. typhimurium*.

### 3.6. Authentic Sample Analysis

To further evaluate the applicability of this proposed method for on-site detection of *S. typhimurium* in milk samples, UHT milk samples spiked with different concentrations of target bacteria we prepared. Appendix A demonstrated that the color of AuNPs changed to purple with naked eye as the concentration of *S. typhimurium* increasing to 2.8 × 10^3^ CFU/mL, and its absorbance ratio (A650/A520) occurred significantly higher. The sensitivity occurred slightly decreased compared to *S. typhimurium* in buffer, owing to the matrix effect of the milk samples and the loss of target cells during centrifugation. Magnetic separation effectively prevented the aggregation interference of AuNPs by proteins and ions in milk samples. The results indicated that our method processes good anti-interference capabilities and accuracy that enabling practical applications.

## 4. Conclusions

In this study, a colorimetric strategy based on aptamer catalyzed hairpin assembly and unmodified AuNPs was successfully developed for on-site detection of *S. typhimurium*. Two aptamers were introduced to specifically identify the target bacteria by magnetic separation and then trigger the Y-CHA circuit. Due to the remarkable optical properties of AuNPs, the change in color from red to blue-gray based on aggregation was easily visualized with the naked eye. Compared with conventional CHA circuit, Y-CHA has 20 times higher sensitivity with the *S. typhimurium* concentration of 2.4 × 10^2^ CFU/mL. Based on the excellent recognition of the aptamer, the assay prevented interference from non-target bacteria and dead *S. typhimurium* cells. In addition, our proposed assay can be extended to the detection of other target bacteria by replacing different aptamer sequences, aiming to pave the way for on-site detection of foodborne pathogens.

## Figures and Tables

**Figure 1 foods-10-02539-f001:**
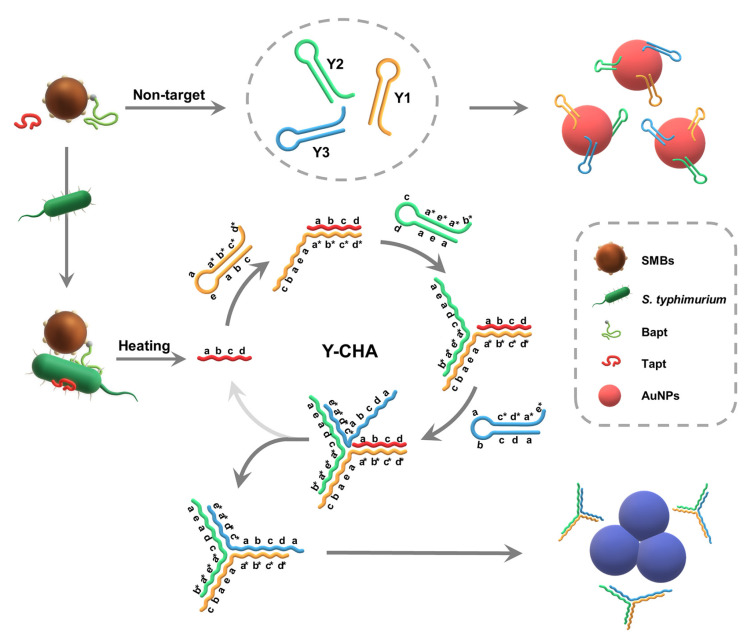
Schematic illustration the on-site colorimetric strategy based on Y-CHA for the detection of *S. typhimurium*.

**Figure 2 foods-10-02539-f002:**
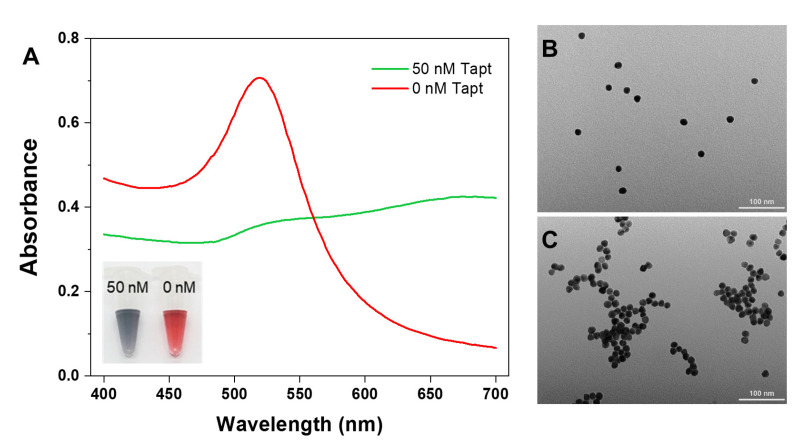
UV–Vis absorption spectra of AuNPs with Y1, Y2, and Y3 in the absence (red line) and presence (green line) of Tapt (**A**). TEM image of AuNPs with Y1, Y2, and Y3 in the absence of Tapt (**B**). TEM image of AuNPs with Y1, Y2, and Y3 in the presence of 50 nM Tapt (**C**). Inset: the photograph of AuNPs with and without Tapt.

**Figure 3 foods-10-02539-f003:**
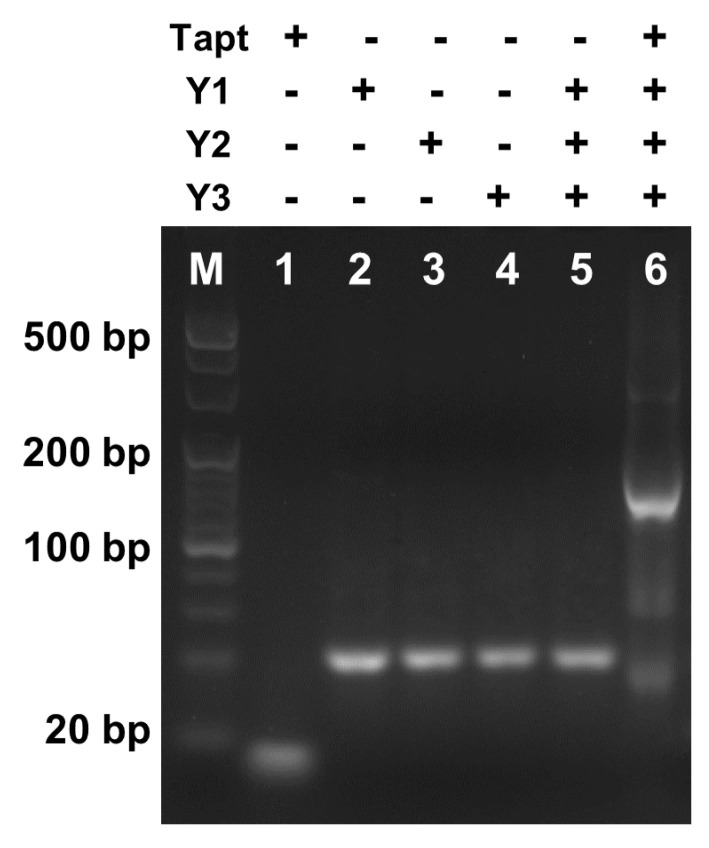
AEG to demonstrate the reaction pathways of the proposed Y-CHA. The “+” and “−” denote the presence and absence of the corresponding DNA components, respectively. The concentration of each hairpin (Y1, Y2, and Y3) initially added is 1 μM, and the concentration of Tapt is 100 nM.

**Figure 4 foods-10-02539-f004:**
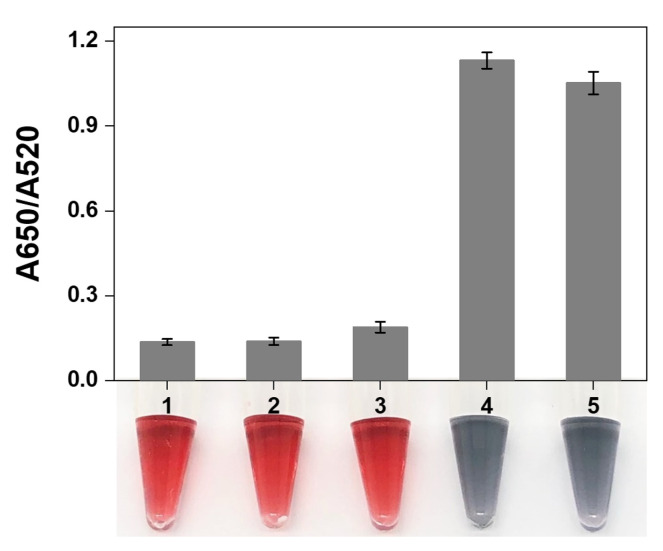
Absorbance ratio (A650/A520) and photograph of AuNPs mixed with various samples. (1) AuNPs, (2) AuNPs + Y1 + Y2 + Y3, (3) AuNPs + Y1 + Y2 + Y3 + NaCl, (4) AuNPs + Tapt + Y1 + Y2 + Y3 + NaCl, and (5) AuNPs + Y1* + Y2* + Y3*+ NaCl.

**Figure 5 foods-10-02539-f005:**
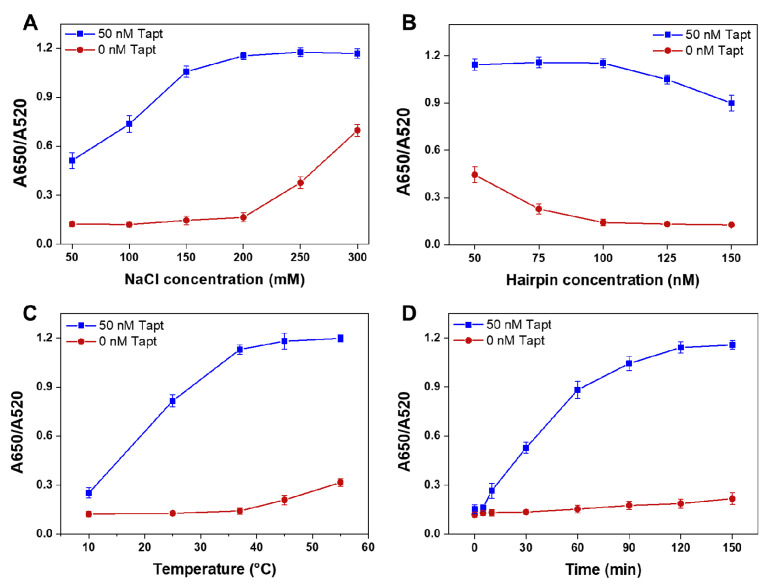
The optimization results of several parameters: concentration of NaCl (**A**), concentration of hairpins (**B**), incubation temperature (**C**), and incubation time (**D**).

**Figure 6 foods-10-02539-f006:**
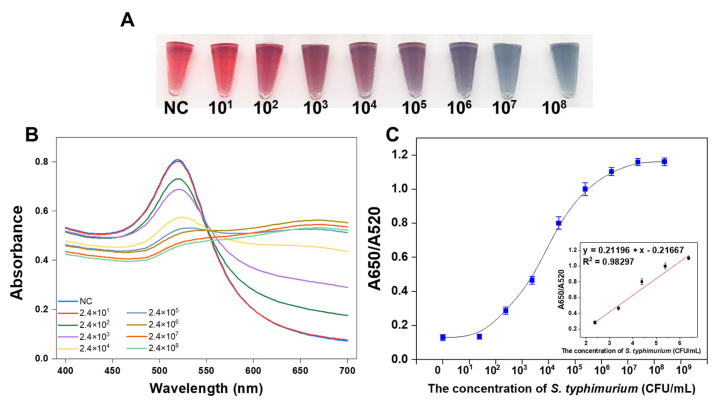
The sensitivity of the colorimetric strategy based on Y-CHA circuit for *S. typhimurium* on-site detection. The photograph of AuNPs for *S. typhimurium* with concentrations ranging from 2.4 × 10^1^ to 2.4 × 10^8^ CFU/mL (**A**). The UV–Vis absorption spectra curve with increasing *S. typhimurium* concentrations (**B**). The linear relationship between A650/A520 and the corresponding *S. typhimurium* concentration (**C**).

**Figure 7 foods-10-02539-f007:**
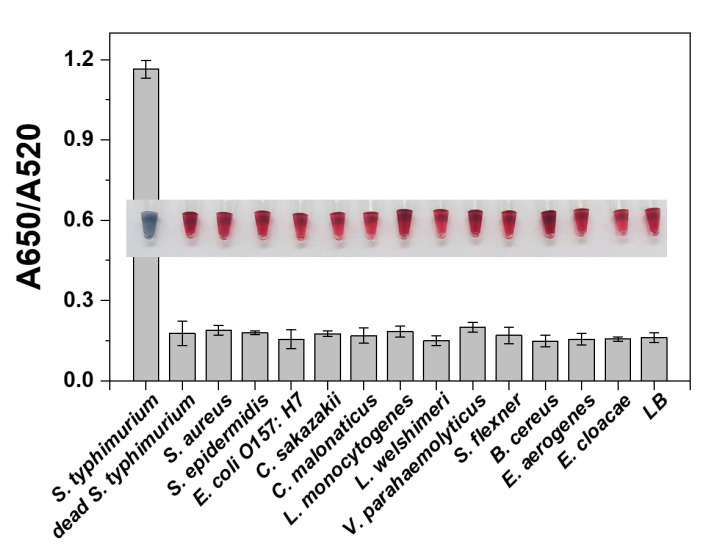
The specificity of the assay was evaluated through comparing several other pathogens and dead *S. typhimurium* with target bacteria. The insert showed the AuNPs visualization colorimetric photo of the corresponding bacterial samples.

## Data Availability

All the sequences of isolated strains and other raw data were submitted to the National Center for Biotechnology Information (NCBI). All the accession numbers have been shown in the materials and methods section.

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
