# Peer review of "A Colorimetric Strategy Based on Aptamer-Catalyzed Hairpin Assembly for the On-Site Detection of Salmonella typhimurium in Milk"

_foods, 2021, doi:10.3390/foods10112539_

Round 1

Reviewer 1 Report

The manuscript entitled "A colorimetric strategy based on aptamer catalyzed hairpin assembly for the on-site detection of Salmonella typhimurium in milk” by Chen et al. concerns the set up and optimization of an aptamer catalyzed hairpin assembly with unmodified gold nanoparticles (AuNPs) to be used as a colorimetric tool for the on-site detection of Salmonella typhimurium in milk.

The topic of the paper is of potential interest for the field of rapid detection of pathogens in food matrices, with particular reference to dairy foods, which are more likely contaminated by this pathogen.

The manuscript needs minor revisions. A particular aspect is the English form. Several sentences are incomplete or lack of verbs, making sometimes the text difficult to understand for the reader.

It would be also useful to discuss why the sensitivity of the method increases from 10^2 to 10^3 when applied to real food samples (i.e., milk): inhibitors? Loss of bacterial cells with centrifugation?

Specific comments:

Line 14: “Here, we set up (or developed) an enzyme free […]”

Line 24: “[…] selectivity even in the presence […]”

Line 24: “has been validate in real milk samples”

Line 34: rely instead of relied

Line 34: “plating on selective culture media and”, instead of “selective plating,”

Line 35: suffer

Line 36: Molecular methods

Lines 41-45: please, rephrase

Line 81: promises

Line 151: Briefly describe the pretreatment process

Line 156: “in triplicate” instead of “three times”

Line 157: S. typhimurium was used as experimental control”

Line 235: optimum conditions

Line 277: increased to around 650 nm

Lines 325-326: please, reformulate

Author Response

It would be also useful to discuss why the sensitivity of the method increases from 10^2 to 10^3 when applied to real food samples (i.e., milk): inhibitors? Loss of bacterial cells with centrifugation?

AU: Thank you for the insightful comments. Although we use some means such as magnetic beads to reduce matrix effects in milk, impurities such as residual proteins in the milk sample would still interfere with the assay, thus reducing the sensitivity of the detection. Besides, the loss of target cells caused by centrifugation during pretreatment can also adversely affect the detection. We have supplemented the discussion of this phenomenon in revised manuscript. (Lines 322-324)

Specific comments:

Line 14: “Here, we set up (or developed) an enzyme free […]”

AU: Thanks for your correction, “we demonstrated an enzyme free” has been corrected as “we set up an enzyme free” in revised manuscript. (Line 14)

Line 24: “[…] selectivity even in the presence […]”

AU: Thanks for your correction, “selectivity whether the presence” has been corrected as “selectivity even in the presence” in revised manuscript. (Line 24)

Line 24: “has been validate in real milk samples”

AU: Thanks for your correction, “has been validate by real milk samples” has been corrected as “has been validate in real milk samples” in revised manuscript. (Line 24)

Line 34: rely instead of relied

AU: Thank you for your correction, we have changed “relied” instead of “rely”. (Line 34)

Line 34: “plating on selective culture media and”, instead of “selective plating,”

AU: Thanks for your correction, we have changed “selective plating,” instead of “plating on selective culture media and”. (Line 34)

Line 35: suffer

AU: The word “suffered” has been corrected to “suffer” in revised manuscript. (Line 35)

Line 36: Molecular methods

AU: Thanks for your correction, we have changed “molecular biology methods” to “molecular methods” in revised manuscript. (Line 36)

Lines 41-45: please, rephrase

AU: Thanks for your careful observation and comments. The sentences have been changed to “In order to overcome the limitations of enzymatic amplification, including complex operations, specific reaction conditions and reaction-time dependent enzymatic activity, a novel signal amplification method, catalytic hairpin assembly (CHA), has been proposed in recent years. CHA is a strategy that depended on self-assembly and disassembly reactions between nucleic acid hairpins to achieve signal amplification, broadening new horizons for on-site detection.” in revised manuscript. (Lines 41-46)

Line 81: promises

AU: Thanks for your correction, “promised” has been corrected to “promises” in revised manuscript. (Line 82)

Line 151: Briefly describe the pretreatment process

AU: Thanks for your kind advice. The pretreatment process was described in the original version of the manuscript, “The 1 mL of overnight activated S. typhimurium cultures […]”. To give a clearer description, we have changed “The pretreatment process of spiked milk samples followed our previous work.” to “With reference to our previous work, the spiked milk samples were pretreated as follows. Briefly,”. (Lines 152-153)

Line 156: “in triplicate” instead of “three times”

AU: Thanks for your correction, “three times” has been corrected to “in triplicate” in revised manuscript. (Line 158)

Line 157: S. typhimurium was used as experimental control”

AU: Thanks for your correction, considering your comment and another reviewer's comment together, this sentence was revised to “S. typhimurium was considered as experimental control” in revised manuscript. (Line 159)

Line 235: optimum conditions

AU: Thanks for your correction, “optimum condition” has been corrected to “optimum conditions” in revised manuscript. (Line 237)

Line 277: increased to around 650 nm

AU: Thanks for your correction, “increased around 650 nm” has been corrected to “increased to around 650 nm” in revised manuscript. (Line 279)

Lines 325-326: please, reformulate

AU: Thanks for your careful observation and comments. We have reformulated the sentence in revised manuscript. The revised sentence is: “In this study, a colorimetric strategy based on aptamer catalyzed hairpin assembly and unmodified AuNPs was successfully developed for on-site detection of S. typhimurium.”. (Lines 329-331)

Reviewer 2 Report

This article aimed to develop an aptamer catalyzed hairpin assembly signal amplification for detection of S. typhimurium. The authors concluded that the on-site strategy developed has excellent selectivity whether the presence of other competitive bacteria, and has been validated by real milk samples.

The article is straightforward, and it contains original information.

This article would be improved if the authors revise the followings:

Line 110, Since it appears first time, spell out “PBS.”

Line 157. Revise to “… was considered as an ….”

Line 182. Revise to “… to prove ….”

Lines 228-229. Consider to revise to “… AuNPs and reduce to some degree the aggregation of AuNPs.”

Line 258. Revise to “A650/A520.”

Author Response

Line 110, Since it appears first time, spell out “PBS.”

AU: Thanks for your kind advice. We have spelled PBS as “phosphate-buffered saline (PBS)”. in revised manuscript. (Line 111)

Line 157. Revise to “… was considered as an ….”

AU: Thanks for your correction, considering your comment and another reviewer's comment together, this sentence was revised to “S. typhimurium was considered as experimental control” in revised manuscript. (Line 159)

Line 182. Revise to “… to prove ….”

AU: Thanks for your correction, “proof” has been changed to “prove” in revised manuscript. (Line 184)

Lines 228-229. Consider to revise to “… AuNPs and reduce to some degree the aggregation of AuNPs.”

AU: Thanks for your correction, “AuNPs and reduce to the degree of aggregation of AuNPs in some degree.” has been changed to “AuNPs and reduce to some degree the aggregation of AuNPs” in revised manuscript. (Lines 230-231)

Line 258. Revise to “A650/A520.”

AU: Thanks for your correction, “A6500/A520” has been changed to “A650/A520” in revised manuscript. (Line 260)